



# Micro-scale and nano-scale strain mapping techniques applied to creep of rocks

Alejandra Quintanilla-Terminel[1], Mark Zimmerman[1], Brian Evans[2], David Kohlstedt[1]

[1]Department of Earth Sciences, University of Minnesota, Minneapolis, MN 55455, USA

[2]Earth, Atmospheric and Planetary Sciences, Massachusetts Institute of Technology, Cambridge, MA, 02139, USA

*Correspondence to*: Alejandra Quintanilla-Terminel (aqt@alum.mit.edu)

**Abstract.** Usually, several deformation mechanisms interact to accommodate plastic deformation. Quantifying the contribution of each to the total strain is necessary to bridge from observations of microstructures to geomechanical descriptions and, thus, is a critical component in the extrapolation from laboratory data to field observations. Here, we describe experimental and computational techniques involved in micro-scale strain mapping (MSSM), which allows strain produced during high-pressure, high-temperature deformation experiments to be tracked with high resolution. MSSM relies on the analysis of relative displacement of initially regularly spaced markers after deformation. We present two lithography techniques used to pattern rock substrates at different scales: photolithography and electron-beam lithography. Further, we discuss the challenges of applying the MSSM technique to samples used in high-temperature and pressure experiments. We applied the MSSM technique to a study of strain partitioning during creep of Carrara marble and grain boundary sliding in San Carlos olivine, synthetic forsterite, and Solnhofen limestone at a confining pressure, *Pc*, of 300 MPa and homologous temperatures, *T/Tm*, of 0.3 to 0.6. The MSSM technique works very well up to temperatures of 700ºC. Experimental developments described here show promising results for higher temperature applications.

## 1 Introduction

Often, during plastic deformation of a crystalline material, strain is accommodated by a combination of several physical processes, which might include ionic diffusion; dislocation glide, climb, and cross slip; mechanical twinning; and grain-boundary sliding [Ashby, 1972; Frost and Ashby, 1982]. Many constitutive models are constructed by prescribing a dominant deformation mechanism and, then, using the physics of that mechanism to establish flow laws, without consideration of the contribution of auxiliary processes. Such simplified flow laws are a useful first step for the extrapolation of laboratory results to larger-scale geomechanical problems, but it is also possible that the relative activity of the various deformation processes might change significantly when extended to natural conditions of rate, mean pressure, chemical environment, and temperature. Thus, quantification of the strain due to each process and determination of the interactions amongst the processes are crucial for establishing robust microphysical models that can be



used to interpret observations of natural micro- and nano-structures and that can describe natural deformation at larger spatial- and time-scales

Quantification of the strain accommodated by each mechanism is not an easy task. Ideally, the microstructural evolution of a polycrystalline material would be monitored continuously as a sample is strained. However, high pressures and temperatures are required when deforming rocks in the laboratory, making microscopic observations of deformation in-situ a significant experimental challenge. Fortunately, even step-wise observations of the amount of strain accommodated by individual intra-crystalline

mechanisms could significantly enhance our understanding of high-temperature creep.

In this paper, we describe a technique for micro-scale strain measurement (MSSM), which permits strain mapping at micrometer and sub-micrometer scales and discuss several observations of strain partitioning in rocks deformed at high pressures and temperatures. To quantify the deformation, the coordinates of the material points (at the micro-scale) must be identified before and after deformation. Then, the strain tensor

can be computed from their relative displacements. We used two lithography techniques, described in Section 2, to print a grid of points onto the polished surface of a rock. After deformation, we computed strain following the formulation described in Section 3. We applied the MSSM technique to four different high-temperature experiments described in Section 4.

**2 Lithographic techniques**

**2.1 Lithography**

The literal meaning of lithography, writing on stones, is very appropriate for our application. More generally, lithography refers to different printing techniques that allow transferring a pattern onto a substrate. Micro- and nano-lithography are widely used in the semi-conductor industry [Mack, 2006] to pattern micro- and nano-metric circuits onto silicon wafers. Applying these techniques to non-conductive,

heterogeneous materials is challenging, but we developed two protocols, using photolithography and electron-beam (e-beam) lithography, to produce patterns on surfaces of Carrara marble, Solnhofen limestone, San Carlos olivine, and synthetic forsterite. The patterns had features with spatial dimensions on the order of one μm down to tens of nm. With some adaptation, these general processes might be used on other geomaterials.

Photolithography and e-beam lithography have unique advantages and disadvantages regarding resolution, fabrication time, and requirements on the sample geometry, that are described in detail below. Both photolithography and electron-beam lithography require special equipment, but such devices are often present in standard clean laboratories used to fabricate solid-state devices. For both techniques, the surface to be patterned has to be polished to a sub-micron roughness. The lithographic processes are analogous to

photographic development in the sense that energy (UV light or an electron beam) is used to transform a





polymer and transfer a motif onto the substrate. As with photographic printing, micro- and nano-lithography involve several processes that need to be adapted to the size of the feature (e.g., grain size) and the substrate, in our case, a rock surface. The protocols described below and illustrated in Figure 1, were optimized for substrates of Carrara marble, Solnhofen limestone, synthetic forsterite, and San Carlos

olivine and the specific patterns were designed for our applications, but these protocols can provide a good starting point for decorating other geomaterials.

The analogy with developing argentic photographs is useful to examine when applying these techniques to rocks. Just as in photographs, the pattern can be over or under exposed. Overexposed features or patterns will appear larger than designed, while those underexposed will be smaller or not appear at all. Some play exists between the exposure time and the development time; ultimately, the lithographer must determine the

optimum match for the intended application. The communities developing and using micro- and nano-fabrication techniques are extremely active. Undoubtedly, new processes and improvements will appear, allowing more innovative applications to geomaterials.

## 2.2 Photolithography

Photolithography was carried out at the MIT Microsystems Technology Laboratory (MTL), where half cylinders of Carrara marble were patterned. First, each half cylinder was polished with a succession of aluminum oxide suspensions down to a 0.5-µm particle size to produce a flat, mirror-like surface. After polishing, the pattern was produced using a series of steps illustrated in Figure 1 and described below:

I.       The polished half cylinders were dried in a convection oven at 130ºC for 20 minutes.

II.      The sample surfaces were manually coated with a light sensitive photoresist, Fujifilm OCG 825 g-line, using a paintbrush; the thickness of the polymer was determined to be ~400 nm using a profilometer. Each half cylinder was then baked for 20 minutes in a convection oven at 90ºC.

III.     The pattern inscribed in a photomask was transferred onto the polymer coating by exposing the coating to ultraviolet light in a mask aligner for 60 s. The exposure time depends on the substrate, the

photoresist, and photoresist thickness. Our photomask was created using computer-aided design (CAD) software and a laser to draw the pattern onto a glass plate covered with chrome and photoresist, followed by a typical etch-back technique [Mack, 2006]. Photomasks are also available from various commercial sources in the microfabrication industry.

IV.     The non-exposed polymer was dissolved using a developer, Fujifilm OCG 934, for 30 to 60 s.

V.       The samples were then very lightly etched with an acidic solution (a 1 mass% dilute solution of HCl for 1 s).



VI.     The surface of the samples was sputtered with a double layer composed of 20-nm of chromium and 30-nm of gold using a plasma sputterer.

VII.     Finally, the protective layer of photoresist was removed with 2-(2-Aminoethoxy)ethanol, N-methyl-2-pyrrolidine at 100ºC for 30 min.

Photolithography is a very flexible patterning technique. Large surfaces can be patterned with a customized design, little limitation exists regarding the geometry of the sample, and a wide selection of metals can be used. For example, the 10-mm by 20-mm patterned area in Figure 2a contains roughly 1.5 million markers spaced every 10-µm and is composed of a double layer of chromium and gold. The polished surface of the outside of a Carrara marble cylinder in Figure 2b is patterned with markers spaced every 50-µm and composed of a double layer of titanium and gold. The main disadvantages of photolithography are related to pattern resolution. Although the resolution of photolithography has been pushed to sub-micrometric scales [Dong et al., 2014], the resolution is limited to a couple of micrometers for non-conventional substrates like rocks and to hundreds of micrometers for patterns on curved surfaces. Furthermore, a photomask is needed and its fabrication adds an additional step in the protocol. More details of the photolithography process and the development of the described protocol are given by Quintanilla-Terminel [2014] and Quintanilla-Terminel and Evans [2016].

### 2.3 Electron-beam lithography

The electron-beam (or e-beam) lithography process was carried out at the Minnesota Nano Center (MNC). The forsterite and San Carlos olivine samples were polished using diamond-lapping film down to a 0.5-µm particle size. The Solnhofen limestone samples were polished with aluminum oxide powder following the same procedure used on the Carrara marble samples. The electron-beam lithography steps are illustrated in Figure 1 and described below:

I.     A wafer of San Carlos olivine, Solnhofen limestone, or synthetic forsterite was initially left in a vacuum oven at 130ºC for 20 min to dry.

II.     The sample was first coated with poly(methyl methacrylate) (PMMA) diluted in 4 vol% chlorobenzene. The coating was done with a precision spin coater, a rotating device that holds the sample via a vacuum jug, spinning at 3000 rpm for 60 s to produce a thickness of the PMMA film of 350-nm (evaluated with ellipsometry). The sample was then baked on a hot plate at 180ºC for 10 min. Afterward, the sample was coated with 15-nm of gold in a plasma sputterer to avoid charging.

III.     The patterning was achieved using a Vistec EBPG5000+ e-beam lithography system. First, the design was created using CAD software. In e-beam lithography, a dose array is typically used in order to find the optimum exposure dose for each material and each resolution. To find the optimum exposure for



San Carlos olivine, forsterite, and Solnhofen limestone, patterns of different resolution were exposed with doses ranging from 300 to 1200 $\mu C/cm^2$ in increments of 50 $\mu C/cm^2$. The optical microscope image in Figure 3a illustrates the results of a dose array performed on a sample of San Carlos olivine. The visible

radial lines forming the spokes of a wheel are 10-$\mu$m thick and were exposed to 900 $\mu C/cm^2$. Inside each rectangle along the radial lines, different doses were tested for grids of lines 50-, 100- and 200-nm wide. Examples of good exposure and over-exposure are shown in the secondary electron micrographs, Figures 3a,b, respectively. Based on these results, we selected optimal doses of 900 $\mu C/cm^2$ for sub-micron features, 750 $\mu C/cm^2$ for micrometer features on San Carlos olivine and synthetic forsterite, and 800 $\mu C/cm^2$ for the

sub-micron features on Solnhofen limestone. A variety of layouts were designed to meet the needs of the type of application; for example, the wheel design (Figure 3a) was developed to evaluate the inelastic strain distribution produced during a torsion experiment.

IV.      The first developing step required dissolution of the gold layer using a solution of KI, routinely

called gold etcher. The sample was first submerged in the KI solution for 30 s, then rinsed in deionized (DI) water for 10 s, and finally rinsed in isopropanol for 10 s. The sample was then developed. Two developing processes were explored. The typical developing solution for PMMA is a 3:1 mixture of isopropanol and methyl isobutyl ketone. The coated sample was submerged in this solution for 60 s. If a higher resolution developing step is required, a mixture of 3:1 isopropanol and DI water can be used. This second mixture

was used with samples that were exposed to about half of the dose previously described in order to avoid over-developing of the features [Yasin et al., 2002]. In any case, the choice of the particular developer, exposure, and developing time should be adapted to the patterned substrate as well as the resolution needed.

V.      The etching process, which strongly depends on the sample, is critical for an improved adhesion of

the metal layer. For olivine, two etching processes were explored, a wet etch using HF diluted to 1 vol% applied for 1 s and a chlorine-gas etch using an Oxford plasma etcher for 1 min. Although the two processes gave similar results, the wet etch was simpler to implement. If the sample cannot be in contact with DI water a gas-etching process would be preferable.

VI.      A 110-nm thick layer of Cr was deposited in a plasma evaporator.

VII.      To dissolve the remaining PMMA, the sample was immersed in 2-(2-aminoethoxy)ethanol, N-methyl-2-pyrrolidine at 120ºC for at least 4 h. Gentle, 1-s bursts of sonication were sometimes necessary to completely remove the PMMA.


VIII.      The resulting patterned surfaces of San Carlos olivine and Solnhofen limestone are illustrated in Figure 4. Notice the difference in scale compared to the patterned marble surface in Figure 2; the images of the patterned Carrara marble in Figure 2 are optical light micrographs, whereas the images of the San





Carlos and Solnhofen limestone in Figure 4 are scanning secondary-electron micrographs. The high
resolution, the sharpness of lines, and the pattern profile can be better appreciated in images of patterns
deposited on San Carlos olivine obtained by atomic-force microscopy (AFM) (Figure 5). It is apparent that
electron-beam lithography permits much higher resolution than photolithography; in theory, nanometer
scales can be achieved [Vieu et al., 2000; Manfrinato et al., 2013]. For geomaterials, a resolution of 20-nm
is easily attained. The pattern is rastered directly onto the sample without the need for a mask, which

facilitates tests of different designs. However, because the rastering process has to be repeated for each
sample, e-beam lithography is much slower and more costly than photolithography. Furthermore, samples
have to fit into the electron-beam lithography system, limiting the geometry of the sample that can be
patterned. In the Vistec EBPG5000+ used at the MNC at the University of Minnesota, the sample has to be
as thin and flat as a silicon wafer (i.e., no more than 1-mm thick), a geometric constraint that has to be

considered when designing deformation experiments.

**2.4 Applying lithography to other geo-materials**

The following list includes some general considerations that may be used to adapt the patterning protocols

to other geo-materials:

1-    Choice of technique: Both photolithography and e-beam lithography can be used for decorating
rocks; a combination of both would be also possible. The main factors to take into account are the size of
the area to be patterned, the design of the pattern (shape and resolution), and the sample geometry.

Photolithography is a good choice when many samples are required, when large areas are to be patterned,
and when the sample geometry is more complicated. For applications requiring a pattern with submicron
features, e-beam lithography will be more appropriate. Furthermore, because e-beam lithography does not
necessitate a mask, it would be faster to implement for single applications.

2-    Polishing step: Samples must be polished to a roughness of less than 0.5-µm; the details of the
technique will depend on the composition, porosity, and grain size of the rock. For electron-beam
lithography, the sample must be flat with less than a 1-µm variation in height for each 1-mm$^2$ region.

3-    Grid design: Either lithographic technique can produce unique patterns specifically designed for a

particular application. For instance, point markers allow strain measurements, but lines may be better suited
to measure offsets between grains. For both techniques, patterns can be designed using commercial CAD
software; we used CleWin 3™ and AutoCad™ as layout editors. Resolution requirements will determine
the choice of lithography technique.




4-     Coating techniques: Choosing between photo- or electron-sensitive polymers and different techniques for coating will depend on the nature of the rock and the geometrical limitations. Spin coaters provide control of polymer thicknesses, but can only be used with flat samples. For more challenging geometries, such as half-cylinders or curved surfaces, a paintbrush or an aerosol spray can be used, but care must be given to ensure that the polymer thickness is relatively uniform from sample to sample.


   5-     Metal required: The two different methods of physical vapor metal deposition described above, evaporation and sputtering, produce different coating characteristics that are important when designing semi-conductors. However, for MSSM patterns, the major concern is achieving a visible pattern that will survive deformation. In general, either technique can be used. In both, there is latitude in the choice of the

material sputtered, typically a metal. Parameters that are important in the choice of metal include adhesion, melting temperature, visibility, and interactions with the rock. For instance, gold does not adhere to carbonates well enough to survive deformation experiments. Therefore, it was necessary to sputter a double layer composed of chromium and gold.

6-     Exposure time/dose and development time: These steps require experimentation, and it is advisable to determine the optimum times from tests of exposure (for photolithography) or dose (for electron-beam lithography).

**3 Strain analysis**


The ultimate goal of using the MSSM technique is to understand which micromechanical processes accommodate strain in a deforming rock and to relate the observed microstructure to the imposed macroscopic deformation conditions. For this purpose, it is necessary to calculate the strain at different scales. The methodologies described here use a Lagrangian description in which strain is calculated by

following a material point before and after deformation [Reddy, 2013; Malvern, 1969].

**3.1 Strain analysis techniques**

Different techniques can be used to track strain based on the position of a material point. Unfortunately, most of them cannot be used for our particular applications at high pressures and temperatures because they

require a continuous description of the position of the material point. We provide here a brief review of some of the strain analysis techniques that have been applied to geomaterials. These techniques compute the strain using the deformation of a random or of a regular pattern on the surface of a rock. The pattern can already be present in the rock, or it can be introduced using different fabrication techniques.





### 3.1.1 Laser speckle photography

Speckle patterns are produced by light scattered from an optically rough surface, and these can be followed
as the body deforms, permitting computation of the full 3-D strain tensor. Laser speckle photography was
initially used to obtain velocity data during hydrodynamic flow [Dudderar and Simpkins, 1977], but has
more recently been used to measure 3-D displacement fields and strain in geo-materials [Barrientos et al.,
2008; Larsson et al., 2004].

**3.1.2 Digital image correlation**

Digital image correlation (DIC) is an excellent tool for extracting displacement measurements [Bruck et al.,
1987; Sutton et al., 2009]. Different algorithms have been developed in order to extract the displacement
field of a deforming body by applying an image correlation between different stages of deformation
[Bornert et al., 2008].  Typically, images need to be similar enough to allow the unequivocal identification
of each individual feature. Thus, DIC is best suited to continuous observations for which computations are
made over very small steps in strain. DIC has been used in a variety of rock mechanics applications: to map
the localization of damage in a heterogeneous carbonate [Dautriat et al., 2011], to quantify the role of
crystal slip and grain-boundary sliding during creep of synthetic halite [Bourcier et al., 2013], and to better
understand creep of ice [Chauve et al., 2015; Grennerat et al., 2012]. Unfortunately, neither laser speckle
photography nor DIC are robust if continuous observation is not possible.

### 3.1.3 Evaluation of a regular grid

Our approach requires identification of a material point before and after deformation, independent of the
amount deformation the material has experienced. We therefore rely on the analysis of an initially regular
grid characterized before and after deformation [Allais et al., 1994; Ghadbeigi et al., 2012; Karimi, 1984;
Moulart et al., 2007; Sharpe, 2008; Wu et al., 2006]. Because our experiments are performed at high
temperature and pressure with a metal jacket surrounding the sample, the grid is introduced in the middle of
the sample, following different variations of a split cylinder assembly [Raleigh, 1965].

The initial reference grid is formed by deposition using either photolithography or e-beam lithography.
Other deposition methods have been used, such as sputtering through a commercial screen [Xu and Evans,
2010] or introducing a metal mesh in between the sample halves [Spiers, 1979]. Recently, Hiraga [2015]
used a focused ion beam to groove lines on samples composed of synthetic forsterite and diopside in order
to track grain displacement and rotation during diffusion creep. However, lithography has the advantages
that higher resolutions can be achieved, that the patterns are more robust, that they survive high-
temperature, high-pressure deformation, and that specific patterns can be custom designed to account for
the grain size of the material or the research questions to be answered. In all the techniques, the main
challenge is to accurately identify each individual marker or line before and after deformation. Various





complications can arise, for instance the markers can be unstable at high temperature, or the split-cylinder surfaces can weld together at high pressures making the recovery of the patterned surface challenging or impossible. Nevertheless, if the identification of individual markers after deformation is accomplished, then

spatial variations in relative displacement can be used to compute the strain field across the reference surface.

Different image processing techniques can be used for evaluating the displacement between identified markers and inverting for strain. Often, some manual input is required to ensure that the marker is correctly identified before and after deformation. Martin et al. [2014] used the described DIC technique on a regular

grid to obtain the strain in a metallic alloy. Biery et al. [2003] and Xu and Evans [2010] used a convolution algorithm to find the markers before and after deformation. Biery et al. [2003] then used a pattern recognition routine or displacement-mapping method to invert for the strain, while Xu and Evans [2010] used the *n*-point algorithm described below. We used a Hough transform algorithm to find the markers and the *n*-point algorithm to invert for the strain [Quintanilla-Terminel and Evans, 2016].

**3.2 The *n*-point analysis**

The *n*-point analysis is an inversion technique that allows the user to probe the strain at different scales. In the following section, we deal with a 2-D strain inversion; strain components in the third dimension can be computed if deformation is isochoric and symmetric about the sample axis. The positions of *n* material points are registered before and after deformation, and the best fit of the deformation gradient tensor

transforming the undeformed to the deformed material lines related to these *n* material points is computed. This inversion assumes that the deformation is homogenous over the area spanned by the *n*-point ensemble. In practice, the technique requires the following steps. (1) Imaging of the region of interest (ROI) before deformation: Based on the size and spacing of the markers, this step can be performed with light or electron beam microscopy. (2) Registration of the coordinates of all markers in the ROI before deformation. (3)

Deformation experiments under pressure and temperature. If the full strain tensor is desired, then conditions should be sufficient to ensure that inelastic strains are isochoric. (4) Imaging of the same ROI after deformation. (4) Registration of the coordinates of all markers in the ROI after deformation. (5) Determination of strain using the *n*-point inversion technique.

The inversion technique assumes homogeneous deformation for all *n* points, and thus, a characteristic

spatial dimension for the local inelastic strain is introduced. In practice, the inversion is realized point by point; for each point, its *n-1* neighbors are located. Thus, when *n* is larger, the computation error is reduced, providing that strain is actually homogenous within the ROI. Thus, a compromise exists between resolution of spatial variations in local strain and errors involved in its calculation. The different *n* configurations and their corresponding strain map and strain distribution along the 1 axis (in the sample reference) are

presented in Figure 6 for a sample exposed to isostatic conditions at $T = 700°C$ and $P_c = 300$ MPa





[Quintanilla-Terminel and Evans, 2016; Xu and Evans, 2010; Bourcier et al., 2013]. This strain map includes all errors, including computational errors, as well as those resulting from experimental artifacts.

To eliminate rigid-body translations, the material displacements are calculated relative to a moving centroid for every set of $n$ points. That is, the coordinates of each point $\mathbf{i}$ before and after deformation, $\mathbf{X}_i$ and $\mathbf{x}_i$ respectively, are referenced to the centroid of the set of $n$ points before and after deformation, $\mathbf{C}_n$ and $\mathbf{c}_n$. The 2-D strain tensor is determined from the relation between material displacement vectors before, $d\mathbf{X}_i = \mathbf{X}_i - \mathbf{C}_n$, and after deformation, $d\mathbf{x}_i = \mathbf{x}_i - \mathbf{c}_n$ using a least-square estimate.

The deformation gradient tensor, $\mathbf{F}$, is determined by calculating the least-square fit for the ensemble of $n$ material vectors, i.e., minimizing the sum of the squares of the difference between the modeled and the measured material lines, $d\mathbf{x}_i^* = \mathbf{F} \cdot d\mathbf{X}$ and $d\mathbf{x}_i$, respectively. Consequently, the tensor, $\mathbf{F}$, describes the homogeneous deformation that best fits the displacement field of $n$ material points. Four different configurations of $n$ points are illustrated in Figure 6a.

### 3.2 From the deformation gradient tensor F to the strain tensor

The deformation gradient tensor $\mathbf{F}$ can be decomposed into a product of a rotation tensor $\mathbf{R}$ and either $\mathbf{U}$, a right-stretch tensor, or $\mathbf{V}$, a left-stretch tensor. $\mathbf{U}$ can be diagonalized following Eq. (1):

$$\mathbf{U} = \mathbf{Q}^T \mathbf{D} \mathbf{Q}. \tag{1}$$

The Hencky strain tensor, $\boldsymbol{\varepsilon}$, is defined in Eq. (2):

$$\boldsymbol{\varepsilon} = \mathbf{Q}^T \ln \mathbf{D} \mathbf{Q}. \tag{2}$$

In the 2-D strain inversion, $\boldsymbol{\varepsilon}$, is a 2 by 2 tensor with three independent components that can be computed in the sample reference as seen in the insert in Figure 9, one along the 1 axis, a second along the 2 axis, and a third shear component. In Figure 9, the three components for an area of Carrara marble deformed to 11% compressive strain at $T = 600^{\circ}C$, $P_c = 300$ MPa and a strain rate of $3 \times 10^{-5} s^{-1}$ are contoured over a map of the grain boundaries of the analyzed area. The geological convention is used in which positive strains correspond to shortening strains and negative strains correspond to lengthening strains.

### 4 Application to creep of rocks

The choice of grid design and patterning technique should reflect the test geometry, physical characteristics of the target rocks, and research goals, including specific hypotheses regarding constitutive behavior. The design choices made for tests on Carrara marble and on San Carlos olivine, synthetic forsterite, and Solnhofen limestone can be used for illustration. In the first case, Carrara marble deforming by power-law creep (cf. Renner and Evans, 2002; Renner et al., 2002), we wished to investigate the partitioning of strain



between different deformation mechanisms, to determine which were dominant, and to identify internal state variables needed for a more accurate flow law [Evans, 2005]. Patterns in this study were made using photolithography. In the second case, we wished to observe microstructures produced during dislocation creep of San Carlos olivine [Hansen et al., 2011; Hirth and Kohlstedt, 2003] and during diffusion creep in

synthetic forsterite [Dillman, 2016] and Solnhofen limestone [Schmid et al., 1977] . Here, our goals included determining the strain contribution of grain-boundary sliding in different creep regimes and testing current constitutive models. For this work, we produced patterns with e-beam lithography.

## 4.1 Grid design and sample geometry

The average grain size of our Carrara marble samples was 130-µm [Xu and Evans, 2010; Quintanilla
Terminel, 2014]; therefore, patterns with micrometer resolution, as formed by photolithography, were optimal. Our designs included circular markers, 2-µm in diameter, with centers spaced at 10-µm intervals, complemented by a printed numbering system. Thus, even though there were about 1.5 million markers, each could be assigned a unique address. Strain was mapped at a scale of 20-µm across the entire sample. For MSSM in samples of San Carlos olivine, synthetic forsterite, and Solnhofen limestone, all of which
have grain sizes ≤10 µm, higher resolution patterns were necessary, and so we used electron-beam lithography to print two intersecting grid patterns. In the first, lines were 500-nm thick, spaced by 4.5-µm; in the second, 200-nm thick, spaced by 2-µm. We chose lines rather than circular markers to emphasize localized offsets at grain boundaries.

Photolithography is a flexible technique that can be used to mark surfaces with varying geometry, including
split-cylinders or cylindrical surfaces (Figures 2a,b, 8a). E-beam lithography has better spatial resolution but limits the surface geometry. As seen in Section 2, the sample has to be flat and not more than 1-mm thick. Two different composite cylinders sample configurations were used to introduce a 1-mm thick patterned sample. In the first, a 1-mm thick disk of San Carlos olivine disk was placed between two short olivine cylinders (Figure 8b). In the second, a rectangular slab, 1-mm thick, of patterned Solnhofen
limestone was inserted between two half cylinders (Figure 8c).

## 4.2 Application of photolithography: Creep of Carrara marble

We used a split cylinder set-up following Raleigh [1965] and Spiers [1979]. Half cylinders of "Lorano Bianco" Carrara marble, a standard material for deformation experiments [Molli and Heilbronner, 1999; Heege et. al., 2002], were polished down to 0.5-µm grit size using aluminum oxide and patterned by
photolithography. Optical micrographs of the surfaces were made before and after deformation. Matlab™ was used for the strain computation. The circular markers were identified using a Hough transform and registered with the embedded coordinate system. The 2-D strain tensor was computed using the $n$-point algorithm. The MSSM technique allowed for an inversion of strain at different scales, up to shortening strains of 36% at a temperature of 700ºC. Notice that strains measured over a ROI at sample-, macro-, and



micro- scales have the same average values, but larger standard deviations as the spatial scale of
measurement decreases (Figure 9).  Maps of the normal strain component in the shortening direction, $\varepsilon_{11}$,
can be used to determine strain accommodation owing to slip along grain boundaries, to twinning, and to
intra-granular deformation. Figure 10 shows samples shortened to 11% at $T$ = 400, 500, 600 and 700ºC and
$P_c$ = 300 MPa; Figure 11 illustrates samples shortened to 11, 22 and 36% at 600ºC. In all experiments, the

means of the local strains agreed remarkably well with those measured during the mechanical tests.
Importantly, the spatial heterogeneity varied with strain and temperature. For more detail, see Quintanilla-
Terminel and Evans [2016].

**4.3 Application of electron-beam lithography: Creep of San Carlos olivine and synthetic forsterite**

The greatest challenge in studying strain distributions during creep in olivine rocks is the separation of the

surfaces after deformation. Because creep tests must be carried out at $T$ > 1000 ºC and at $P_c$ = 300 MPa, the
metal of the grid and the rock surface strongly adhere. Nonetheless, the results described here are
promising and suggest that, with further work, strain distributions during grain boundary sliding might be
successfully measured (Figures 12-13).

**4.3.1 Compression experiments on San Carlos olivine at 300 MPa and 1150ºC**

We used electron-beam lithography to pattern mono-phase, polycrystalline samples prepared from powders
of San Carlos olivine, $Fo_{90}$. First, samples were fabricated by cold pressing powders into a nickel can,
followed by hot-pressing at $P_c$ = 300 MPa and $T$ = 1250°C for 3 h [Hansen et al., 2011]. A 1-mm thick slice
was cut from the hot-pressed cylinder and one side was ground and polished with a final step using lapping
films with 0.5-μm diamond grit. The olivine wafers were then patterned following the protocol described in

Section 2. The rest of the cylinder was cut in half, one end polished, and the three pieces assembled as
shown in Figure 8b. This composite sample was deformed at $P_c$ = 300 MPa, $T$ = 1150 ºC, and a constant
strain rate of $1 \times 10^{-5}$ s$^{-1}$ to 15% shortening strain in a gas-medium high-pressure apparatus [Hansen et al.,
2011]. The strength at steady state of the composite, 245 MPa, is in excellent agreement with that
determined from earlier tests of intact cylinders [Hansen et al., 2011].

Recovering the grid after deformation was more challenging because of the high temperatures and normal
loads on the marked surface. We experimented with different metals and separation techniques. Sputtering
a window shape of a high melting temperature metal such as tungsten on the facing sample (a technique
similar to the one used for Carrara marble) was unsuccessful, and the two faces still could not be separated.
Inserting a thin (0.01-mm), ring-shaped, nickel foil between the gridded surface and the facing surface was

still not sufficient, but the gridded surface was successfully recovered if a full nickel foil was placed
between both surfaces. Secondary electron micrographs of grids with lines 200-nm and 1-μm thick show
offsets of 0.5 to 1-μm along the grain boundaries (Figures 12a, b). On a recovered area (Figure 13a), the
strain was inverted using the *n*-point algorithm described above; the component $\varepsilon_{22}$ is mapped in Figure 13b.





Note that the gridded surface is perpendicular to the shortening axis, compared to the Carrara marble tests in which the gridded surface was parallel to the shortening axis (Figures 10-11). Because the size of the recovered area is small, we cannot draw meaningful conclusions regarding the partitioning between intragranular strain and grain boundary sliding, but the technique offers further opportunities for success.

### 4.3.2 Compression experiments on forsterite at 1 atm and 1100ºC

Samples of synthetic forsterite were synthetized following Koizumi et al. [2010]. After gridding, they were deformed at $T$ = 1250ºC and $P_c$ = 1 atm. [Dillman, 2016] with the grid on an outer surface of a sample with a square-cross section. The tests used a grid 500-nm thick and were designed to compare with previous measurements of creep [Dillman, 2016] using a line offset caused by grain-boundary sliding [Langdon, 2006]. Grid recovery for these samples was much easier because deformation occurred under atmospheric pressure. We used an AFM to measure the vertical displacement of grains and to map the deformed grid
(Figure 14). The state of the initially regular grid after deformation highlights the deformation occurring along grain boundaries. 2-D offsets of ~500 nm are evident. More details on the 1 atm. compression experiments on synthetic forsterite can be found in Dillman [2016].

### 4.3.3 Creep of fine-grained Solnhofen limestone

Three-piece samples of Solnhofen limestone were prepared (Figure 8c), gridded using e-beam lithography, and deformed in compression in gas-medium apparatus [Paterson, 1990] at $T$ = 700ºC and $P_c$ = 300 MPa to a shortening strain of 9% at a strain rate of $3x10^{-4} s^{-1}$. The patterned surfaces could be separated without the addition of a metal foil, and the deformed pattern was easily found using electron microscopy. In Figure 15 with two secondary electron micrographs of the same area before and after deformation, the deformed lines and grid are clearly visible, however the grain structure is not. Additional characterization, perhaps imaging
with electron beam scattered diffraction, will be necessary to evaluate the contribution of grain boundary sliding to the total strain. Nonetheless, this application demonstrates that the deformation of the lines and grid is visible and that the electron-beam lithography provides the required resolution to quantify the strain at the granular level.

### 5. Conclusion

Patterning using photolithography and e-beam lithography can provide maps of strain calculated over spatial scales of 10-1 μm, respectively. The experimental preparation is technologically demanding and labor intensive, but MSSM is a unique characterization tool that yields a detailed description of strain accommodation within intra- and inter-granular regions. The experimental protocols described here can be adapted to many other rocks, and the pattern can be designed for specific purposes. The MSSM technique
has the potential to improve the interpretation of microstructural observations of rocks deformed in nature



and in the lab, to constrain the partitioning of strain amongst several deformation mechanisms, and to improve constitutive modeling of creep mechanisms in the Earth.

*Acknowledgements*

Matej Pec is acknowledged for his help acquiring the SEM pictures. Amanda Dillman is acknowledged for providing the data related to the forsterite sample and for her help acquiring the AFM data. The lithography techniques were developed at the Minnesota Nano Center (MNC) of the University of Minnesota and at the MIT Micro Technology Laboratories (MTL). AQT thanks Bryan Cord at the MNC and Kurt Broderick at the MTL for their guidance through the fabrication processes. We benefitted from enriching discussions

with Matej Pec, Amanda Dillman and William Durham. Support through NSF Grants EAR-1520647 (UMN) and 145122 (MIT) is gratefully acknowledged.

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



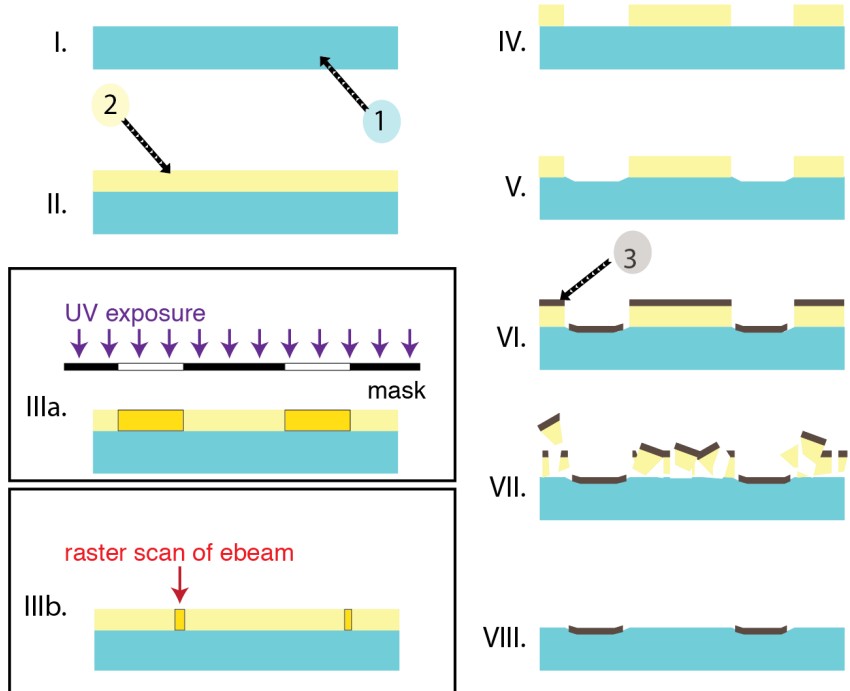

**Figure 1: Lithographic process: 1- rock, 2- photoresist or PMMA, 3-metal layer (sputtered or evaporated). Each step is described in more detail in the text: I- Sample preparation, II- Coating with a polymer (photoresist for photolithography, PMMA for e-beam lithography), III- Exposure with UV rays for photolithography (IIIa) or with an electron beam for e-beam lithography (IIIb), IV- Development of the coating, V- Etching, VI- Metal deposition, VII- Dissolution of the remaining polymer, VIII- Patterned sample.**



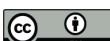



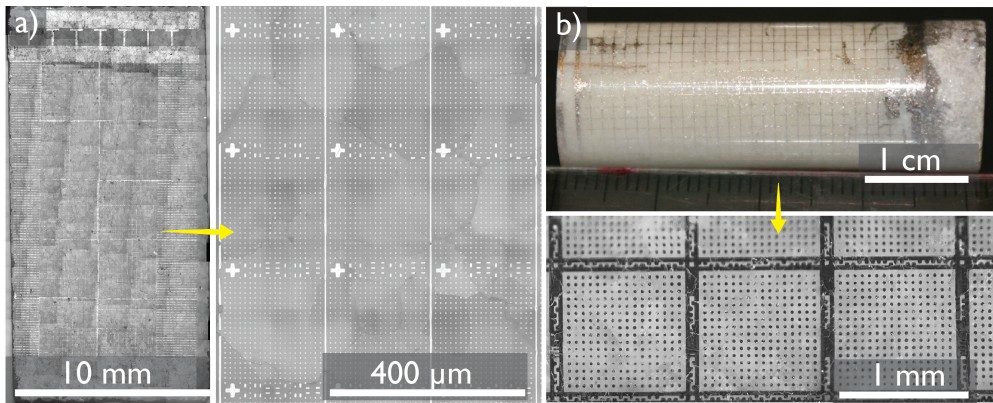

**Figure 2: Patterned Carrara marble using photolithography. a) The optical light micrograph on the left shows**
**the patterned half-cylinder with more than 1.5 million markers, while the micrograph on the right presents a**
**zoomed-in view of an area revealing the 2-µm wide markers with centers spaced every 10-µm as well as the**
**embedded numbering system that allow us to locate each marker before and after deformation. b) The**
**photography on top illustrates the structure of a patterned cylinder, while the optical light micrograph on the**
**bottom show a closer view of the patterned curved surface revealing 20-µm wide markers with centers spaced**
**every 50-µm.**







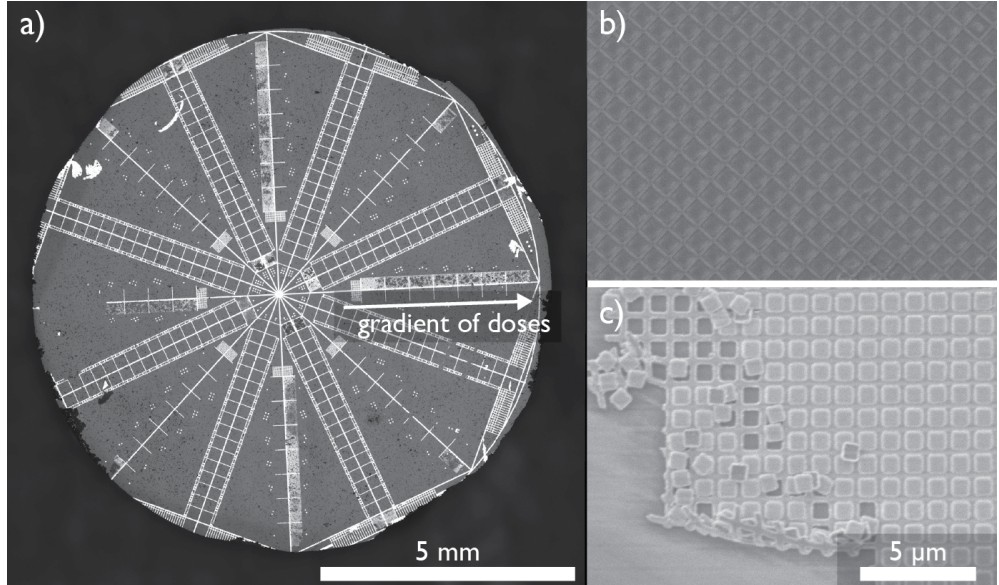

**Figure 3: a) Light micrograph of a patterned San Carlos olivine crystal. Electron beam lithography was used with varying doses along the radial lines. All radial lines (10-µm thick and exposed with a 900µC/cm$^2$ dose are visible), inside each rectangle different doses were tested: an example of a well exposed and an over-exposed area are seen in (b) and (c) respectively. In the over-exposed area, the pattern appears blurred, and causes the rounding of the corners in the grid. An under-exposed area will typically show no signs of the pattern. The dose test allowed identification of the optimum dose for achieving the desired resolution: 700 µC/cm$^2$ for 200-nm wide lines.**






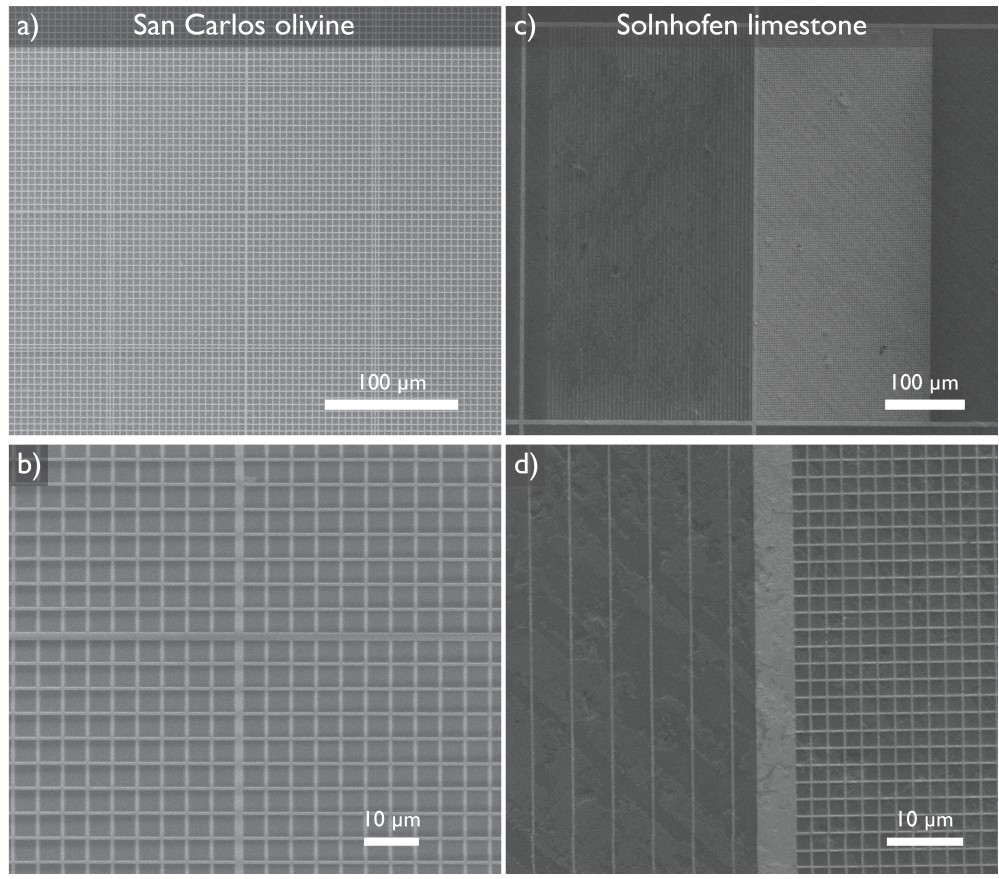

**Figure 4: Secondary electron images of patterned San Carlos olivine (a and b) and Solnhofen limestone (c and d).**
**The grid in the San Carlos olivine sample is composed of 500-μm lines spaced every 4.5-μm (a), a zoom in is seen**
**in (b). For Solnhofen limestone, we combined 200-nm straight lines with a regular grid of 200-μm lines spaced**
**every 2-μm (c), a zoom into both areas is seen in (d).**






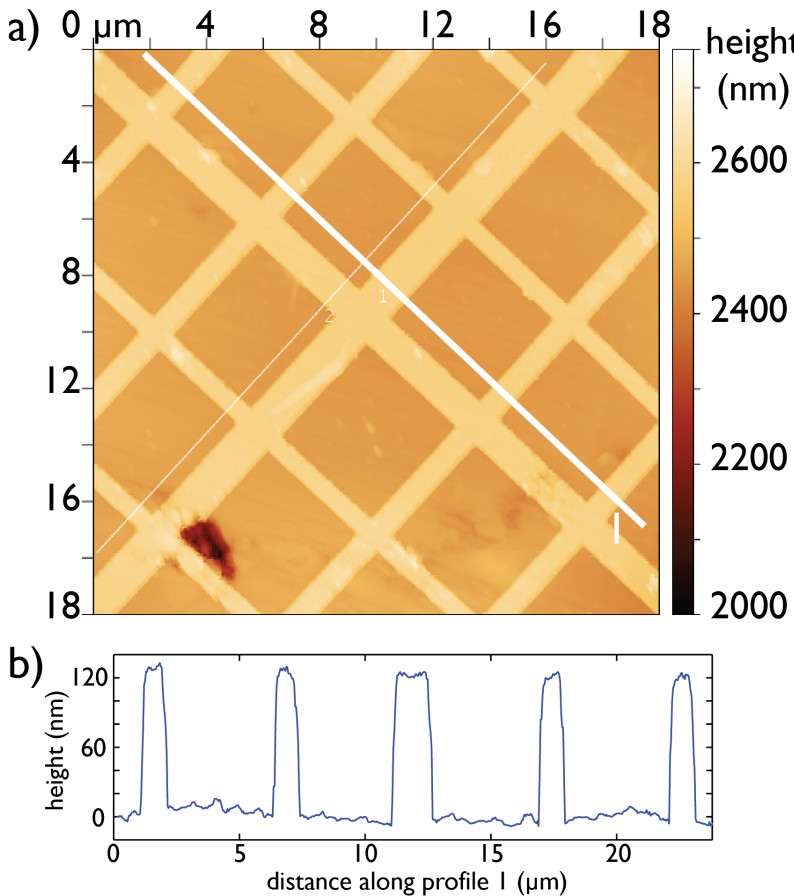

Figure 5: Atomic force micrographs of patterned San Carlos olivine (a) and height profile (b) of the surface along the profile 1. The 500-nm wide chromium lines are spaced every 4.5 μm and are 110 nm high. The sharpness of the lines can be appreciated on the AFM data.





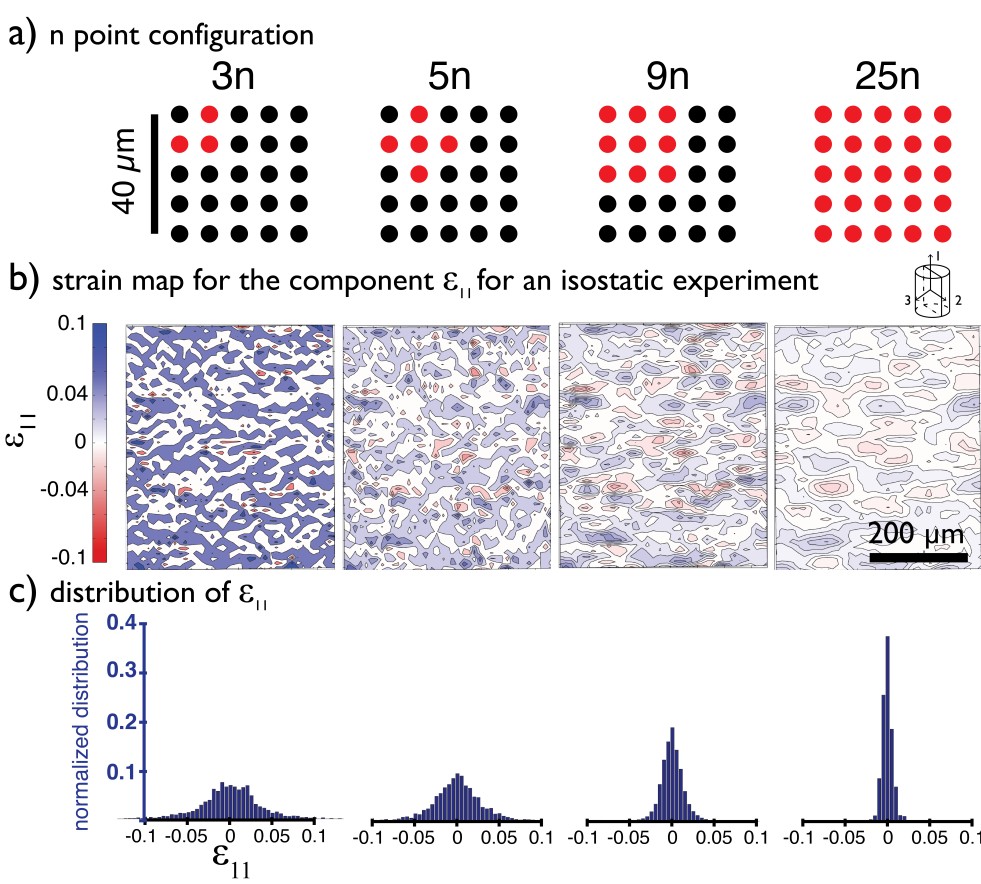


**Figure 6: a)** *n* **point configurations for** *n***=3, 5, 9 and 25. b) Corresponding strain maps and strain distributions for strain along the axis 1 in the sample reference (see insert for reference frame) for a Carrara marble cylinder isostatically annealed at** *T* **= 700ºC and** *P*$_c$ **= 300 MPa for 3 hours. c) Corresponding strain distribution for each strain map, it can be observed that the mean is always 0 but the spread of the distribution is smaller the larger** *n*

**is.**




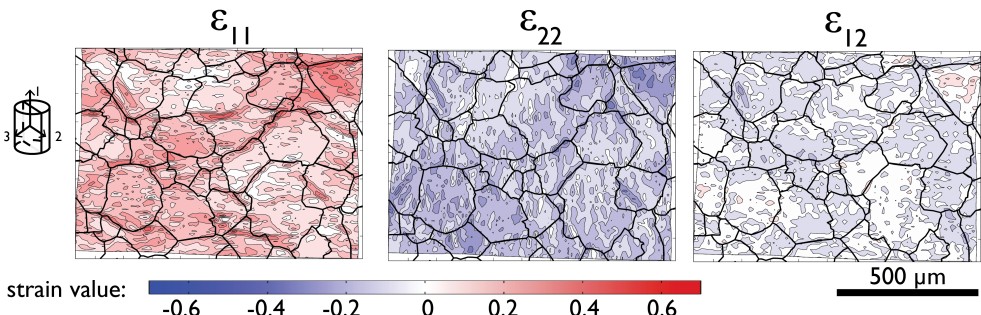

Figure 7: Strain maps for the three components of the 2-D strain tensor for a sample of Carrara marble deformed at $T$ = 600ºC, $P_c$ = 300 MPa to 11% shortening strain at $3x10^{-5}s^{-1}$. The strain tensor was computed

using the $n$ point technique with $n$ = 9. The grain boundaries are overlaid over the strain maps. Positive strains correspond to shortening strains.





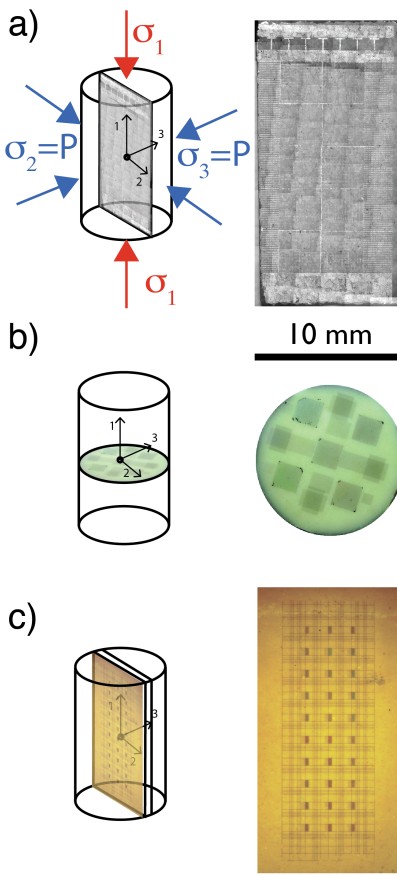

**Figure 8: Experimental set-up for studying creep of Carrara marble, San Carlos olivine, and fine grained Solnhofen limestone at high temperatures and $P_c$ = 300 MPa in a Paterson Instruments gas apparatus. In (a) the composite half-cylinder is composed of two pieces, one is gridded using photolithography, the other is polished and sputtered with a window shape metal layer. Figures (b) and (c) show the composite set-ups used when the patterning involved e-beam lithography. San Carlos olivine was deformed using set-up (b): a patterned 1-mm thick disk was introduced between two short olivine samples. Solnhofen limestone was deformed using the set-up in (c) : a patterned 1-mm thick rectangular slab was introduced in-between two half-cylinders.**






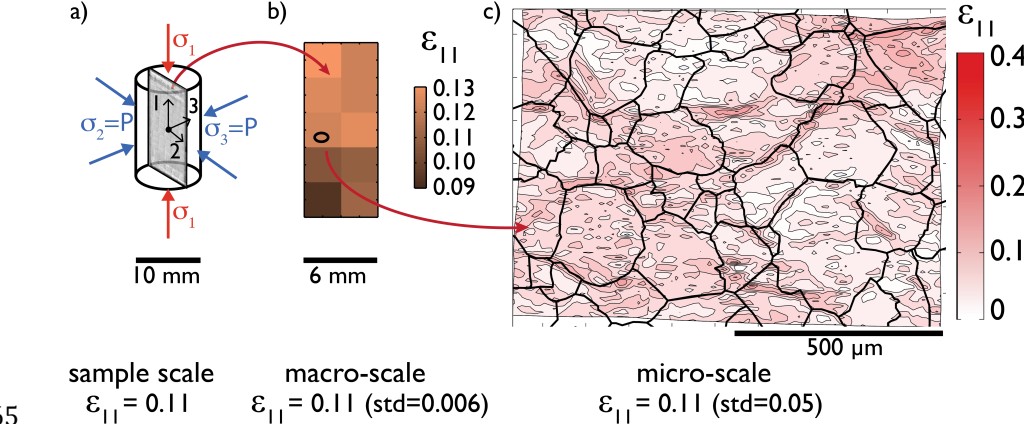


**Figure 9:** Strain maps at different scales for a Carrara marble split cylinder deformed at $T = 600°C$, $P_c = 300$ MPa, to 11% shortening strain at $3 \times 10^{-5}$ s$^{-1}$ (a). The component along the shortening direction of the strain tensor, $\varepsilon_{11}$, was computed at different scales. In (b), $\varepsilon_{11}$ was computed over the whole sample with a resolution of 3-mm (macro-scale). In (c), $\varepsilon_{11}$ was computed over a smaller section with a 20-µm resolution (micro-scale) and is

mapped with the overlaying grain boundaries. Positive strains correspond to shortening strains.





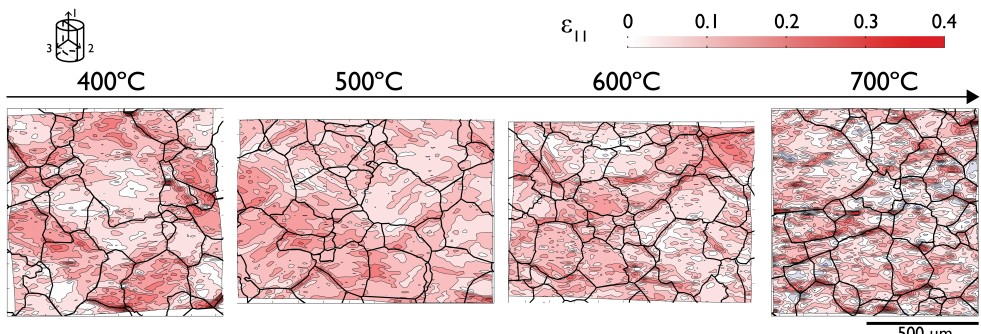

**Figure 10: Strain maps for $\varepsilon_{11}$ computed with a *n* point technique (with *n*=9) for samples of Carrara marble deformed at $T$ = 400°C, 500°C, 600°C and 700°C, $P_c$ = 300 MPa, $3 \times 10^{-5} s^{-1}$ to 11% shortening strain. The strain component along the shortening direction is mapped with the overlaying grain boundaries. Positive strains correspond to shortening strains. Note the concentration of strain along grain boundaries and intra-granular features as *T* increases.**






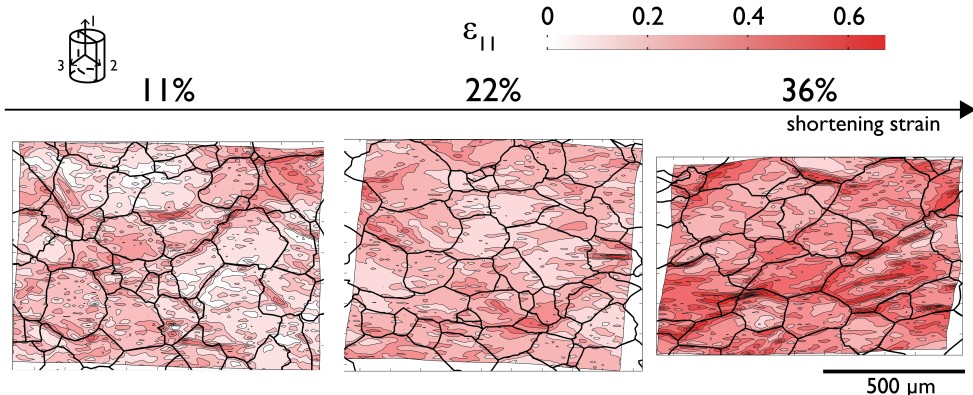

**Figure 11: Strain maps for $\varepsilon_{11}$ computed with a _n_ point technique (with _n_=9) for samples of Carrara marble deformed at _T_ = 600°C, $P_c$ = 300 MPa, $3 \times 10^{-5} s^{-1}$ to 11%, 22% and 36% shortening strain. The strain component along the shortening direction is mapped with the overlaying grain boundaries. Positive strains correspond to shortening strains.**








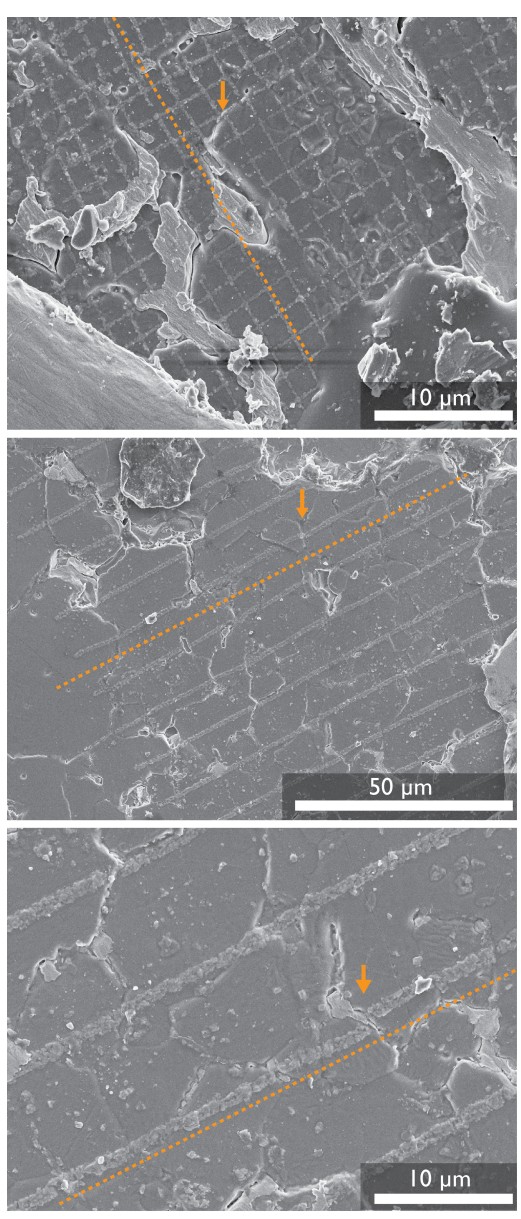

**Figure 12: Secondary electron images of a gridded fine-grained San Carlos olivine after deformation at $T$ = 1100ºC, $P_c$ = 300 MPa to 5% shortening at $1x10^{-5}s^{-1}$. The deformed lines show clear offsets at grain boundaries. The dotted orange lines put in evidence the deformation of the lines in the grid. The orange arrows show grain boundary displacement put in evidence by the grid.**






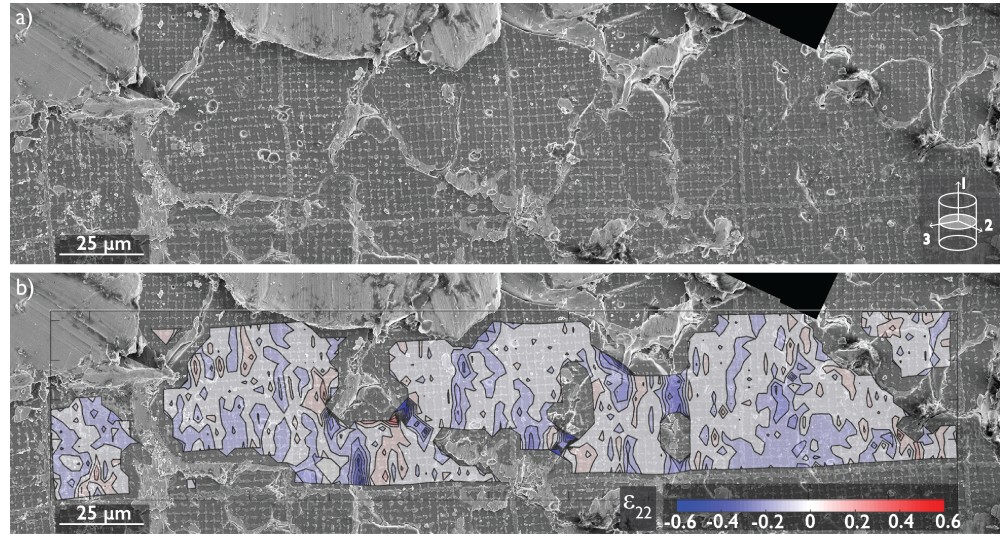

**Figure 13: Strain inversion on recovered grid in fine-grained sample of San Carlos olivine deformed at $T$ = 1100ºC, $P_c$ = 300 MPa to 5% deformation at 1x10$^{-5}$s$^{-1}$. a) Secondary electron image of the recovered area and b) superposed strain map of $\varepsilon_{22}$, a strain tensor perpendicular to the shortening direction.**








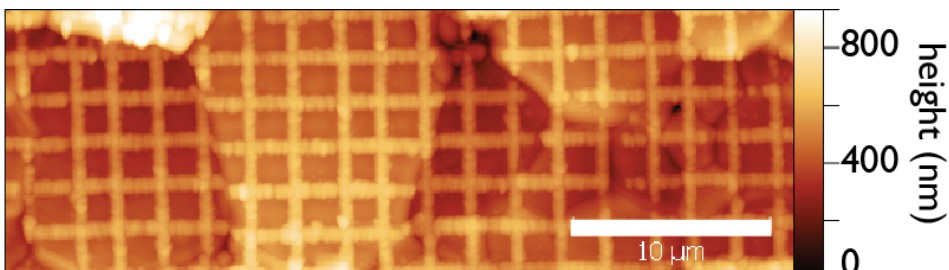

**Figure 14: Atomic force micrograph of deformed grid in synthetic forsterite deformed at $T$ = 1110ºC, $P_c$ = 1 atm to 5% shortening. The chromium lines are about 110-nm thick and are clearly visible in the height map. The grain structure is also clearly visible and both the height difference and the offset of the lines show evidence of grain displacement. Amanda Dillman performed the deformation experiment and the AFM observations for this sample.**







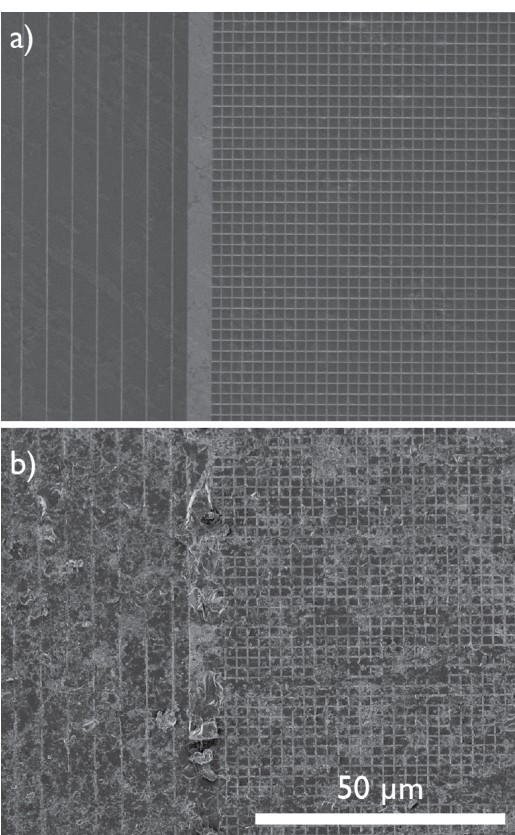

**Figure 15: Secondary electron micrographs of a patterned Solnhofen limestone sample (a) before deformation and (b) after deformation at $T = 700°C$, $P_c = 300$ MPa, to 9% shortening strain at $3 \times 10^{-4} s^{-1}$. Before deformation the pattern consisted on 200-nm width lines spaced every 5-μm and of a grid of 200-nm lines spaced every 2-μm. The lines were made of 110-nm thick chromium. After deformation, the lines are clearly deformed but the grain structure is not apparent.**
