# Peer review of "Micro-scale and nano-scale strain mapping techniques applied to creep of rocks"

_Solid Earth, 2017_

## Referee Comment (RC1) · Anonymous Referee #1 · 13 Apr 2017

The manuscript by Quintanilla-Terminel et al. focuses on the techniques to quantify small-scale strain during deformation of geo-materials. The authors present numerous examples of imposing surface markers using lithographical techniques to the samples that undergo high-temperature deformation. They are succeeded in detecting nano- to micro-scale strain in calcite and olivine aggregates from deformation of the markers. Such a fine scale strain analysis indeed helps our extrapolation of experimental results to nature which are quite different in scales of time, grain size, pressure and so on. The paper is very technical and many details are supplied that will help others who want to follow the complex technique of strain mapping. I recommend publishing with minor modifications.

[Figure]

Some comments

25. "ionic" should be "atomic"?, since the material is not necessarily ionic materials here.

50. I recommend the author to reconstruct this section of 2. I think the title of section of 2.4 can be as "General procedures of lithography to geomaterials". If that is the case, then I recommend the author to bring this section before the section of 2.2. The contents of 2.2 and 2.3 are examples of applications of the techniques that are explained at the beginning of this section 2.

Perhaps, the title of "2.1 Lithography" can be removed. The content of 2.1 is a general overview of lithograhic techniques such that 2.1 can start from "Photolithography".

---

## Referee Comment (RC2) · L. Hansen (Referee) · 22 Apr 2017

It was a pleasure to read and review "Micro-scale and nano-scale strain mapping techniques applied to creep of rocks" by Quintanilla-Terminel, Zimmerman, Evans, and Kohlstedt. This manuscript outlines state-of-the-art techniques for analyzing the distribution of strain at length scales below the grain size in crystalline solids. As pointed out by the authors, these techniques are invaluable for determining the microphysical mechanisms of deformation and evaluating constitutive models used to extrapolate predictions of mechanical behavior from the laboratory to geological conditions. Small-scale strain mapping has been developed and used extensively in the metallurgical literature, but these techniques have seen little application to geological materials because of the need for extreme conditions during deformation (a notable exception that should be mentioned is ice, e.g., Grennerat et al., Acta Materialia, 2012). In the current manuscript, the authors have explored and documented the laboratory procedures and computational techniques allowing small-scale strain mapping to be used at high temperatures and confining pressures. This is certainly a worthwhile contribution suitable for publication in Solid Earth pending revision.

I have two general comments followed by some minor specific and technical comments.

First, it would be useful for me to have the benefits of the "Regular Grid" laid out more clearly. The text suggests that the technique requires images of the same region before and after deformation. If this is strictly true, then it needs to be stated more clearly and upfront in the manuscript. In addition, if before and after images are necessary, then it is a bit unclear to me why a regular grid is beneficial over random markers like those used in digital image correlation (DIC). There are several points in the text discussing DIC for which I've made specific comments below. The authors make the point that DIC requires continuous in situ imaging, but I think this phrasing is misleading. Most in situ techniques pause the actuators when imaging because the deformation actuators introduce too much mechanical vibration. Thus, there is a "before" and "after" image taken, similar to the technique described here. The similarities extend further. DIC use particle tracking or cross correlation to get displacement fields describing the motion of the particles from one image to the next. Much of the discussion of the Regular Grid technique sounds like a similar process. The centroids of a group of particles are compared in "before" and "after" images to determine the displacement of the centroid from one image to the next. Is there an additional benefit of the regular grid that I'm missing?

I do wonder if there is a way to implement the Regular Grid method without a "before" image. If true, I would see this as a major benefit of the method. Because the grid is, by definition, regular, you already know the original grid spacing and therefore could simply compare the deformed grid to a synthetic reference. Is something like this

possible?

Second, I think the manuscript is lacking a discussion of a critical topic, the resolution of the techniques developed. Resolution is discussed in qualitative terms, primarily focussing on the need for finer grid spacings to evaluate smaller deformations. However, it would be valuable to quantify achievable resolutions for different experimental setups. For instance, what is the precision in locating the markers or in locating the centroid of a group of markers? What is the minimum strain that is resolvable with these techniques? What is the measured strain if a sample is put together but not deformed (this would provide an estimate of the noise floor for the strain measurement)? How does the strain resolution vary between reflected light and SEM? How does the strain resolution depend on n (in the n-point technique)? Addressing some of these questions would be very helpful in evaluating whether these techniques are appropriate for specific future applications.

Again, I think the manuscript is an important contribution overall and expect any revisions resulting from my general, specific, and technical comments to be relatively minor. I'm highly supportive of publication of a revised version in Solid Earth.

Sincerely,

Lars Hansen, Department of Earth Sciences, University of Oxford

Specific comments:

Line 26: Perhaps it is worth noting what is mean by "dominant". Are the authors referring to the process that contributes the most to the total strain? Or are they referring to the process that limits the strain rate? Also, what is meant by "deformation mechanism"? Does this refer to the individual strain producing processes listed in the first sentence of the paragraph? Or are the authors referring to combinations of processes, such as dislocation creep, which might consist of dislocations moving by glide and climb.

Line 36: This seems like a bit of a red herring. Yes, in situ observation is great, but I'm not really sure what it gains you other than the ease of not having to move a sample between apparatus at each strain step. Most in situ experiments are stepwise in nature anyway since imaging can't be carried out while actuators are moving.

Line 66: Where is the polymer in this context? How is it being transformed? This is detailed below, but it is a bit confusing at this point in the text when the method has not yet been explained.

Line 235: I think it is worth stating here what "continuous description" means and why it is necessary.

Line 240: It is worth noting that this technique requires images of the same region before and after deformation. Also, isn't this method just DIC but using laser speckles as the markers? If so, perhaps section 3.1.1 should come after 3.1.2.

Line 255: Again, the authors need to describe why continuous observation is a necessity.

Lines 308 to 317: The text here discusses measurement of strain first and then discusses measurement of the deformation gradient tensor. However, isn't this out of order considering the next section suggests F is determined first and then decomposed into strains and rotations?

Line 324: This section describes the analysis for 2-D strain inversion. However, earlier in the section (Line 288), the authors suggest 3-D strain can be calculated assuming the deformation is isochoric and symmetric about the sample axis. A little more description about this process would be useful. To assume the strain is symmetric about the sample axis, doesn't one of the 2-D principal strains need to be aligned with the sample axis? It seems unlikely this would be the case in the majority of situations. What seems simpler to me is to assume the 2-D principal strains are also principal strains in 3-D.

Line 372: It is not immediately clear from the figure how the measured strain can be partitioned into strain due to slip on boundaries, twinning, and intragranular deformation. A quick explanation would be useful, or at least a reference to Quintanilla-Terminel and Evans (2016).

Line 425: It is not clear from the text why EBSD (note that "beam" should be "backscatter") is necessary to evaluate the strain from grain-boundary sliding. Is this just because grain boundaries need to be identified, or is there another explanation?

Technical comments:

Line 34: How about "temporal" instead of "time" to parallel "spatial"?

Line 568: Change "photography" to "photograph".

Figure 5: What is profile 2? I don't see it referenced anywhere. Also, there is a stray vertical white line in the lower right corner. I'm not sure what its purpose is.

---

## Author Comment (AC1) · 26 May 2017

The manuscript by Quintanilla-Terminel et al. focuses on the techniques to quantify small-scale strain during deformation of geo-materials. The authors present numerous examples of imposing surface markers using lithographical techniques to the samples that undergo high-temperature deformation. They are succeeded in detecting nano- to micro-scale strain in calcite and olivine aggregates from deformation of the markers. Such a fine scale strain analysis indeed helps our extrapolation of experimental results to nature which are quite different in scales of time, grain size, pressure and so on. The paper is very technical and many details are supplied that will help others who want to follow the complex technique of strain mapping. I recommend publishing with minor modifications.

25. "ionic" should be "atomic"?, since the material is not necessarily ionic materials here.
Changed in text.

50. I recommend the author to reconstruct this section of 2. I think the title of section of 2.4 can be as "General procedures of lithography to geomaterials". If that is the case, then I recommend the author to bring this section before the section of 2.2.
The contents of 2.2 and 2.3 are examples of applications of the techniques that are explained at the beginning of this section 2.
Perhaps, the title of "2.1 Lithography" can be removed. The content of 2.1 is a general overview of lithograhic techniques such that 2.1 can start from "Photolithography".
We have re-written this section. Thank you for the suggestion!

---

## Author Comment (AC2) · 26 May 2017

Thank you for your helpful review and comments. We have addressed each point and changed the manuscript accordingly. We re-wrote the part describing the need for a regular grid and added a paragraph concerning the resolution of the technique. We attach here a pdf with your review and comments and the answer to each of them.

Please also note the supplement to this comment:
http://www.solid-earth-discuss.net/se-2017-27/se-2017-27-AC2-supplement.pdf

[Figure]

**Supplement:**

**Reply to L. Hansen review (detailed)**

**L. Hansen (Referee)**

lars.hansen@earth.ox.ac.uk

It was a pleasure to read and review "Micro-scale and nano-scale strain mapping techniques applied to creep of rocks" by Quintanilla-Terminel, Zimmerman, Evans, and Kohlstedt. This manuscript outlines state-of-the-art techniques for analyzing the distribution of strain at length scales below the grain size in crystalline solids. As pointed out by the authors, these techniques are invaluable for determining the microphysical mechanisms of deformation and evaluating constitutive models used to extrapolate predictions of mechanical behavior from the laboratory to geological conditions. Small-scale strain mapping has been developed and used extensively in the metallurgical literature, but these techniques have seen little application to geological materials be cause of the need for extreme conditions during deformation (a notable exception that should be mentioned is ice, e.g., Grennerat et al., Acta Materialia, 2012). In the current manuscript, the authors have explored and documented the laboratory procedures and computational techniques allowing small-scale strain mapping to be used at high temperatures and confining pressures. This is certainly a worthwhile contribution suitable for publication in Solid Earth pending revision.

I have two general comments followed by some minor specific and technical comments. First, it would be useful for me to have the benefits of the "Regular Grid" laid out more clearly. The text suggests that the technique requires images of the same region before and after deformation. If this is strictly true, then it needs to be stated more clearly andupfront in the manuscript. In addition, if before and after images are necessary, then itis a bit unclear to me why a regular grid is beneficial over random markers like thoseused in digital image correlation (DIC). There are several points in the text discussing DIC for which I've made specific comments below.  The authors make the point that DIC requires continuous in situ imaging, but I think this phrasing is misleading. Most in situ techniques pause the actuators when imaging because the deformation actuators introduce too much mechanical vibration.  Thus, there is a "before" and "after" image taken, similar to the technique described here. The similarities extend further. DIC use particle tracking or cross correlation to get displacement fields describing the motionof the particles from one image to the next.  Much of the discussion of the RegularGrid technique sounds like a similar process. The centroids of a group of particles are compared in "before" and "after" images to determine the displacement of the centroid from one image to the next.  Is there an additional benefit of the regular grid that I'm missing?

The DIC method requires that the imaged areas are similar enough between deformation steps so that each area can be correctly identified. If the areas are too different the

algorithm can mismatch them and a lot of manual input is necessary. The similarity of each area is a condition that is not fulfilled in our experiments (where the high pressure and high temperature step renders the post-deformation image too different from the pre-deformation). The use of a regular grid allows us to calculate the strain without worrying about misidentifications.

With the use of the regular grid one could indeed calculate the strain without imaging the sample before. However, it is beneficial to have imaged the before stage since it provides additional information, for instance, one can locate grain boundaries and identify the markers that belong to a particular grain.

I do wonder if there is a way to implement the Regular Grid method without a "before" image. If true, I would see this as a major benefit of the method. Because the grid is, by definition, regular, you already know the original grid spacing and therefore could simply compare the deformed grid to a synthetic reference. Is something like this possible?

It is a good point, it is possible: the spacing in the grid is defined and if there are not irregularities in the pattern the strain can be computed without imaging the sample before deformation. In fact, the strain map for the olivine samples was computed by assuming a regular grid without the "before" positions. We did image the area before to establish that the pattern did not have any irregularity. For the Carrara marble application, we did identify the markers before deformation, this step was useful for identifying the markers belonging to each grain (see Quintanilla-Terminel, 2014).

Second, I think the manuscript is lacking a discussion of a critical topic, the resolution of the techniques developed. Resolution is discussed in qualitative terms, primarily focussing on the need for finer grid spacings to evaluate smaller deformations. However, it would be valuable to quantify achievable resolutions for different experimental setups. For instance, what is the precision in locating the markers or in locating the centroid of a group of markers? What is the minimum strain that is resolvable with these techniques? What is the measured strain if a sample is put together but not deformed (this would provide an estimate of the noise floor for the strain measurement)? How does the strain resolution vary between reflected light and SEM? How does the strain resolution depend on n (in the n-point technique)? Addressing some of these questions would be very helpful in evaluating whether these techniques are appropriate for specific future applications.

Thank you for the suggestion. We have added a paragraph dealing with the resolution of the technique.

Again, I think the manuscript is an important contribution overall and expect any revisions resulting from my general, specific, and technical comments to be relatively minor. I'm highly supportive of publication of a revised version in Solid Earth.

**Specific comments:**

Line 26: Perhaps it is worth noting what is mean by "dominant". Are the authors referring to the process that contributes the most to the total strain? Or are they referring

to the process that limits the strain rate?  Also, what is meant by "deformation mechanism"?  Does this refer to the individual strain producing processes listed in the first sentence of the paragraph? Or are the authors referring to combinations of processes, such as dislocation creep, which might consist of dislocations moving by glide and climb.

It is a good point, we have modified the text and hopefully it is more clear. In fact, we meant the processes used in the micro-physical models used to fit the experimentally obtained flow laws. Both dominant or limiting processes could be used, depending on the micro-physical model established, but this discussion is not the focus of this paper.

Line 36: This seems like a bit of a red herring. Yes, in situ observation is great, but I'm not really sure what it gains you other than the ease of not having to move a sample between apparatus at each strain step. Most in situ experiments are stepwise in nature anyway since imaging can't be carried out while actuators are moving.

We disagree in this point. While our stepwise observations provide us with a lot of information, many additional questions would greatly benefit from an in-situ observation. For instance, how do twins propagate? What is their speed?  Furthermore, the depressurization and cooling steps affect the deformation and some mechanisms might not be able to be identified correctly because of this.

Line 66:  Where is the polymer in this context?  How is it being transformed?  This is detailed below, but it is a bit confusing at this point in the text when the method has not yet been explained.

We have changed the text for more clarity.

Line 235: I think it is worth stating here what "continuous description" means and why it is necessary.
Line 240:  It is worth noting that this technique requires images of the same region before and after deformation. Also, isn't this method just DIC but using laser speckles as the markers? If so, perhaps section 3.1.1 should come after 3.1.2.
Line 255: Again, the authors need to describe why continuous observation is a necessity.

The laser speckle technique relies on the interference between two coherent light sources, one coming from the surface that is deforming. The strain is evaluated from the fringes resulting from such interference. It is not the same as DIC that requires an image of the surface. The strain is here evaluated from the cross-correlation of the same block of pixels before and after deformation. We have deleted the part on laser speckle to avoid confusion.
The block of pixels that are being cross-correlated need to be similar enough for DIC to work, otherwise areas can be misidentified leading to spurious strain calculations. Continuous observation is indeed not necessary but the surface needs to remain similar enough. We have changed the text for clarity.

Lines 308 to 317:  The text here discusses measurement of strain first and then discusses measurement of the deformation gradient tensor. However, isn't this out of order considering the next section suggests F is determined first and then decomposed

into strains and rotations?

The deformation gradient tensor is computed from the positions of the markers, and the strain tensor is then computed form it. The strain tensor is indeed computed from the deformation gradient tensor. We have changed the text for clarity.

Line 324: This section describes the analysis for 2-D strain inversion. However, earlier in the section (Line 288), the authors suggest 3-D strain can be calculated assuming the deformation is isochoric and symmetric about the sample axis. A little more description about this process would be useful. To assume the strain is symmetric about the sample axis, doesn't one of the 2-D principal strains need to be aligned with the sample axis? It seems unlikely this would be the case in the majority of situations. What seems simpler to me is to assume the 2-D principal strains are also principal strains in 3-D.

In order to compute the full 3D tensor, we would need the 3 coordinates of each point, however, some deformation geometries allow for some simplifications. The assumptions taken in a split cylinder geometry are described in Quintanilla-Terminel & Evans 2016.

Line 372: It is not immediately clear from the figure how the measured strain can be partitioned into strain due to slip on boundaries, twinning, and intragranular deformation. A quick explanation would be useful, or at least a reference to Quintanilla-Terminel and Evans (2016).

Thank you for the suggestion, an explanation was added to the text.

Line 425: It is not clear from the text why EBSD (note that "beam" should be "backscatter") is necessary to evaluate the strain from grain-boundary sliding. Is this just because grain boundaries need to be identified, or is there another explanation?

Yes, it is just to get the grain boundaries in this case. The typo on EBSD has been changed.

Technical comments:
Line 34: How about "temporal" instead of "time" to parallel "spatial"?
Changed in text.
Line 568: Change "photography" to "photograph".
Changed in text.
Figure 5: What is profile 2? I don't see it referenced anywhere. Also, there is a stray vertical white line in the lower right corner. I'm not sure what its purpose is.
Profile 2 was pretty much identical to profile 1 so we did not include it. We think the stray vertical may be a publishing artifact because it is not apparent in our original image.